# The effects of cloud-aerosol-interaction complexity on simulations of presummer rainfall over southern China

Kalli Furtado[1], Paul Field[1], Yali Luo[2], Tianjun Zhou[3], and Adrian Hill[1]

[1]Met Office, Exeter, UK
[2]Chinese Academy of Meteorological Sciences, Beijing, China
[3]Institute of Atmospheric Physics, Beijing, China

**Correspondence:** Yali Luo (ylluo@cma.gov.cn)

**Abstract.** Convection-permitting simulations are used to understand the effects of cloud-aerosol interactions on a case of heavy rainfall over south China. The simulations are evaluated using radar observations from the South China Monsoon Rainfall Experiment and remotely sensed estimates of precipitation, clouds and radiation. We focus on the effects of complexity in cloud-aerosol interactions, especially depletion and transport of aerosol material by clouds. In particular, simulations with aerosol concentrations held constant are compared with a fully cloud-aerosol-interacting system to investigate the effects of two-way coupling between aerosols and clouds on a line of organised-deep convection. It is shown that cloud processing of aerosols can change the vertical structure of the storm by using up aerosols within the core of line, thereby maintaining a relatively clean environment which propagates with the heaviest rainfall. This induces changes in the statistics of surface rainfall, with a cleaner environment being associated with less intense but more frequent rainfall. These effects are shown to be related to a shortening of the timescale for converting cloud-droplets to rain as the aerosol-number concentration is decreased. The simulations are compared to satellite-derived estimates of surface rainfall, condensed-water path and the outgoing flux of short-wave radiation. Simulations for fewer aerosol particles out-perform the more polluted simulations for surface rainfall, but give poorer representations of top-of-atmosphere radiation.

## 1   Introduction

Physical models of clouds and aerosol microphysics are complex components of atmospheric simulators and, because they are fundamental to the Earth's energy and hydrological cycles, they are a large source of uncertainty in predictions across a wide range of time-scales: from weather forecasts and seasonal predictions, out to climate projections.

Complexity in microphysics schemes arises from the number of processes being modelled and how many prognostic variables are used. Simple single-moment schemes use hydrometeor mass as the prognostic variable for each of cloud droplets, rain and ice. More complex schemes differentiate between sub-species of hydrometeor (graupel, hail, cloud ice and snow) or employ more than one prognostic for each species. Greater complexity improves physical realism but raises the computational expense and it is not obvious where the balance between cost and benefit lies. Moreover, the relative importance of the different mechanisms by which aerosols affect clouds and precipitation are themselves uncertain (Tao , 2012). Although the basic hypotheses that cloud-droplet number concentrations can alter the brightness, longevities and amounts of clouds are well-established (Twomey , 1977; Albrecht , 1989; Rosenfeld et al , 2008), how these processes combine to determine the responses of systems of clouds has been found to depend on both the system under consideration (Kaufman et al , 2005; Rosenfeld et al , 2008) and the model being used (Hill et al , 2015; Johnson , 2015). For deep-convective clouds in particular, uncertainty abounds because increased droplet numbers are associated with both increased and decreased rainfall in manner that appears sensitive to several factors including: ice formation, the large-scale environment and history of the evolving aerosol-cloud system (Khain et al , 2008; Miltenberger et al , 2018).

A range of complexities are also involved in aerosol schemes. Speciated models treat the population of aerosols as composed of physically distinct species, for example salts of sulphuric acid or sodium, organic- and inorganic-carbon compounds. Each species is distributed across a set of size ranges (modes), the contents of which are described by prognostic variables. Such descriptions clearly necessitate a large number of prognostics.

Cloud-aerosol interactions add further complexity because the sophistication with which these are modelled can itself be varied. In their simplest form, the effect of aerosols on the number concentration, $N_c$, of droplets or ice crystals, involves parametrizing $N_c$ as a function of the number concentration, $N_a$, of the aerosol particles:

$$N_c = N_{\mathrm{act}}^c(N_a, \dots), \tag{1}$$

where the *ellipsis* $(\dots)$ represents dependencies on other parameters such as atmospheric-state variables. A formulation such as Eq. 1 is suitable for use in a single-moment microphysics scheme, and the aerosol concentration can be a diagnostic parameter or can evolve dynamically via an aerosol scheme.

In microphysics schemes with prognostic cloud-number concentrations, Eq. 1 is modified to take the form of a source term in the dynamical equation for $N_c$ which specifies an increment to the number of cloud particles when the conditions for activation are met.

An advantage of double-moment schemes is that microphysical processes can feedback on aerosol concentration. With a single-moment scheme, the only permitted feedbacks are *sink* terms: aerosols can be depleted during activation but there is no number-conserving way of accounting for in-cloud processing of aerosol material and hence no way of determining how many aerosols are returned to the air when clouds evaporate. Such models are therefore inherently physically inconsistent: in reality, activated aerosol must be recycled during evaporation and are not simply 'lost' , or removed, from the system. Double-moment schemes, because they include a budget for cloud-number concentrations, allow for two-way coupling between aerosols and

clouds. In this case, mechanical processing of aerosol inside cloud particles can be modelled explicitly, so particles which acted as cloud nuclei can agglomerate in solution and be re-deposited into the air as larger particles.

The above considerations suggest two levels of aerosol-cloud complexity:

1. *One-way coupling of a fixed population of soluble aerosols.* A parameterization is used for activation of cloud droplets from a fixed population of water-soluble aerosols, but aerosol-number concentrations remain constant[1]. The sole route to nucleation of ice-crystals is via aerosol-independent freezing of cloud droplets.

2. *Two-way coupled clouds and soluble aerosols.* Aerosols are depleted during activation and aerosol mass is carried through clouds by sedimentation of hydrometeors and undergoes mechanical processing during collision-coalescence. When hydrometeors evaporate, the air is re-populated with processed aerosols. Because effective cloud nuclei are by definition highly deliquescent, the number concentration of aerosol residuals will equal the number of evaporated hydrometeors. Hence the residuals will typically be less numerous but larger in size than the initial aerosols. In the simplest case, ice nucleation remains independent of aerosols (although, in principle, it could be dealt with in an analogous way).

Two-way coupling represents the *minimum level of complexity in model physics* required to represent depletion of aerosol during activation. We note that there exists a lower complexity, double-moment system in which aerosols are depleted by activation but are not recycled through clouds. It is not our intention to investigate such models here because they suffer from similar physical inconsistencies to single-moment schemes, and hence do not give a physically meaningful representation of cloud-aerosol coupling. In this paper we will compare the *commonly used* fixed-aerosol assumption to the minimum-complexity, two-way coupling ((2, above); with the aim of understanding what new phenomena –if any– arise from consistently coupling clouds to aerosols, and whether these provide any benefits for model performance. By considering fixed-aerosol experiments with a range of aerosol concentrations, we identify candidate mechanism for the differences between the one- and two-way coupled simulations.

Two-way coupled schemes were initially developed using detailed size-resolved descriptions of aerosols and clouds (Feingold and Kreidenweis , 2002). However, a more minimal requirement is prognostic variables for the mass and number concentrations of interstitial aerosols and the mass of aerosol present inside each hydrometeor species. (In general, surface emissions of particulates are also required but here we assume that the time-scales for such processes are slow compared to the forecast duration.) Such schemes have recently been incorporated into bulk models of cloud microphysics, but their testing for applications in weather and climate models is still in its relative infancy. In this regard, the study by Miltenberger et al (2018) is of particular relevance to this paper, because it employs the same modeling system. Those authors used the Cloud-AeroSol Interacting Microphysics (CASIM) in regional simulations with the Met Office Unified Model (UM) to investigate the effects of aerosol-cloud interactions on convective clouds over the south-west peninsula of the United Kingdom. They showed that increasing the number of aerosol particles increased the number of convective cells but decreased the mean-cell size. They found that higher concentrations of aerosols suppressed rainfall when the convection was relatively disorganised, but enhanced

---

[1]To prevent over-production of cloud-droplets, activation does not occur if the parametrization diagnoses fewer activated droplets than already exist in a grid box. When activation does occur it increases the droplet number to the diagnosed number of condensation nuclei.

rainfall when the convection was organised along low-level shear lines (a phenomena that they attributed historical effects of antecedent rainfall on the available moisture as the clouds evolved). In addition, they found that when aerosol processing was included, a simulation with a given initial aerosol concentration tended to behave analogously to a non-processing scheme with a *lower* aerosol concentration (see in particular their Figures 5 and 6). In terms of convective storms globally, Miltenberger et al (2018) considered relatively low-intensity, small-scale rainfall events. To improve understanding of the affects of aerosols

on forecasts of extremes of rainfall, there is a need to test aerosol-interacting models of bulk-cloud microphysics on heavier-rainfall cases. The monsoon regions present an ideal setting for this because of the frequent occurrence of globally significant rainfall extremes. In this paper, we apply the CASIM microphysics to a squall line which, in contrast to previously studied cases, was close to 1000-km across, produced clouds over 15 km in depth, and sustained heavy rainfall rates for a period of several days.

Fan et al (2012) also examined the effects of *fixed-verses-processed* aerosol concentrations on deep-convective clouds over eastern China using a size-resolving ("bin") microphysics scheme. They showed that with both fixed and dynamic aerosols, more polluted conditions were associated with decreased rain water content and suppressed (temporally delayed) rainfall. Their results showed that fixed aerosols exaggerated the responses of cloud and rain to aerosol perturbations, because aerosol processing provided a negative feedback on cloud-droplet number which was not captured if the aerosols were fixed. Interest-

ingly, the opposite phenomena, i.e., increased sensitivity of rainfall to aerosol when processing was permitted, was reported by Miltenberger et al (2018).

In this paper we will use the CASIM microphysics to study the sensitivity of a squall-line of organised-deep convection that occurred over southern China in May 2016. The region receives the majority of its annual rainfall at this time of year, mainly from warm-sector convection. The synoptic situation studied here is one of the most frequently occurring modes of

convective organisation in the region (Huang , 2018), hence understanding whether cloud-aerosol interactions can affect the rainfall produced by such systems may have implications for improving predictions of regional rainfall extremes (Luo et al , 2017; Zhang et al , 2018). Moreover, since future generations of operational weather forecast models will be able to include two-way coupling of clouds are aerosols, it is our intention to contribute evidence regarding the role of two-way coupling in short-range predictions of precipitation extremes. Such information will be useful to developers of forecasting systems

when deciding if increasing the complexity of cloud-aerosol coupling is operationally valuable. The impact of aerosol-cloud interactions on model performance will be evaluated by using ground based radar measurements and satellite-remote sensing.

## 2    Methods

In this section we describe the model experiments and the observations used to evaluate the simulations.

### 2.1    Model description

This study uses a convection-permitting configuration of the Met Office Unified Model. A description of the model set-up can be found in Furtado et al (2018), together with a detailed description of the non-aerosol components of the CASIM mi-

**Table 1.** Descriptions of the model experiments.

| Experiment | description |
| --- | --- |
| 5e$x$F, $x \in \{5,\dots,8\}$ | fixed-aerosol with number $N_a = 5 \times 10^5, \dots, 5 \times 10^8$ m$^{-3}$, and mass $\rho_a = 1.5 \times 10^{-9}$ kg m$^{-3}$ |
| 1s0dP | aerosol-processing with one soluble species intialised with $N_a = 5 \times 10^7$ m$^{-3}$, $\rho_a = 1.5 \times 10^{-9}$ kg m$^{-3}$ |

crophysics scheme (see also Grosvenor et al (2017)). In this paper we use the double-moment configuration of CASIM, in which five species of hydrometeor (cloud, rain, ice, snow and graupel) are described by prognostic mass and number concentrations. Conversion of cloud-droplets to rain is parametrized following Khairoutdinov and Kogan (2000). Other inter-species transfers of mass and number are handled as accretion processes with bulk-collection kernels determined by the fallspeeds and collision-cross sections of the sedimenting particles. The aerosol concentrations are either treated as prescribed constants

throughout the domain, or are initialised with a spatial homogeneous value that is then allowed to evolve via two-way coupling of the clouds and aerosols. The coupling between the cloud and aerosol fields is described in Miltenberger et al (2018), but the salient features of the coupling are as follows: firstly, aerosols are removed from the air when cloud-droplets are activated (using the parametrization developed by Shipway (2015)); secondly an additional prognostic variable for in-cloud aerosol mass is co-advected with the hydrometeors so that it is transported conservatively through clouds; finally, when cloud particles

evaporate, the in-cloud soluble material is returned to the air with a number concentration equal to the number of evaporated hydrometeors. Hence, when aerosols are redeposited during evaporation, their mean size usually exceeds that of the previously activated aerosols (because collision-coalescence gives rain drops that are fewer in number than the cloud droplets from which they develop). This implies that aerosol that were activated as "accumulation"-mode-sized particles can be converted to larger ("coarse") mode particles during evaporation. In section 3 we will compare a simulation with processing of aerosol particles

by clouds to simulations with fixed-aerosol concentrations. A nomenclature for referring to these experiments is established in Table 1. In general, we use the notation $N_a$F for a fixed-aerosol experiment with number concentration $N_a$, and refer to the two-way coupled experiment as "1s0dP", where 'P' is for *Processing* and the prefix indicates that there is one species of soluble aerosol particles (1s) no insoluble ("dust") aerosol species (0d). For the fixed-aerosol experiments we consider reductions of $N_a$ in decades, from approximately $500/\text{cm}^3$. This range is selected to span the range of concentrations generated in

1s0dP. For reference, some of the plots also include an unrealistically clean, 'limiting' case with $N_a = 5 \times 10^5$. The lack of dust aerosols means that there are no prognostics for aerosol particles that can nucleate ice crystals in the simulation (other than liquid-water droplets). This does not imply that the only pathway to producing ice is homogeneous freezing: heterogeneous freezing is included via a temperature-dependent parametrization for the effects of immersion freezing which specifies the fraction of droplets that become ice nuclei (Cooper , 1986). This fraction is a function only of temperature and is independent

of the number of interstitial aerosol particles. Ice-crystal number concentration can be indirectly affected by the number of aerosol particles, because the number of cloud droplets can affect the number of ice crystals.

## 2.2 Case overview: 19-21 May 2016 (SR1)

The simulations are evaluated against ground-based radar observations from the Southern China Monsoon Rainfall Experiment (SCMREX; Luo et al (2017)) and satellite-derived estimates of top-of-atmosphere (TOA) radiative fluxes and surface rainfall rates. The case chosen ("SR1") is an example of organised warm-sector convection that occurred between the 19 May and 21 May 2016 over southern China. A baroclinic environment with a significant amount of large-scale control led to a squall-line of organised deep-convection that propagated along the south coast of China, in Guangdong and Jiangxi provinces. The squall moved eastwards, over a twelve hour period, bringing heavy rain to the coast of Fujian province by 18 UTC 20 May (02 BJT 21 May), before eventually traveling out over the South China Sea.

## 2.3 Observations and metrics

To evaluated the model simulations we use surface-rainfall retrievals from the Global Precipitation Measurement (GPM) mission, radar observations from the SCMREX campaign and broadband radiant fluxes from the Clouds and the Earth Radiant Eneregy System (CERES) instrument on NASA's *Aqua*. When we compare models and observations, the analysis is conducted after re-griding all the datasets to a fixed latitude-longitude grid with a grid-spacing corresponding to the lowest-resolution data set included in each comparison. A brief description of the observations and some relevant uncertainty estimates are as follows.

### 2.3.1 GPM

We use post-real-time rainfall estimates from GPM missions' Integrated MultisatelliE Retrievals for GPM (IMERG) dataset. IMERG is a calibrated, multi-sensor retrieval that provides 30-minute, $0.1° \times 0.1°$, estimates of precipitation. The performance of IMERG for east Asia is known to be good in comparison to surface rainfall measurements: Ning et al (2016) reported biases of less than 0.1 mm/day in daily-mean rainfall over 20-month comparison, and showed that IMERG captured both the amount and occurrence frequency of heavy rainfall during that period; Wang et al (2017) investigated cases of extreme rain and showed that IMERG is accurate to within 10 percent over the 20-60 mm/h range.

### 2.3.2 SCMREX-radar measurements

Measurements of radar reflectivity-factor, $Z$, are obtained from a S-band radar located in Guangzhou at $(113.35°E, 23.00°N)$. The maximum range of the radar is approximately 200 km and volume-scans containing 9 elevations are available at intervals of 6 minutes. The azimuthal-resolution of the scans is 1 degree and there are 900 equispaced radial-gates. For comparision to the simulated radar reflectivities, each azimuthal scan is interpolated onto a fixed-height grid as descibed in Furtado et al (2018).

### 2.3.3 CERES Single-scanner footprint (CERES-SSF)

Top-of-atmosphere longwave (LW) and shortwave (SW) radiative fluxes from the CERES scanning-broadband radiometers are used to provide SW and LW fluxes at 20-km spatial resolution (Wielicki et al , 1996). We use the Aqua edition 3A Single-scanner Footprint (SSF) data which gives calibrated radiances in at 0.3-5 $\mu$m, and 8-12 $\mu$ with estimated uncertainties of 5 W/m$^2$ and 2 W/m$^2$ (Loeb et al , 2007).

## 3 Results and Discussion

### 3.1 Effects and mechanisms of cloud-aerosol interactions

Figure 1 shows observations and simulations at 06 UTC on 19 May (corresponding to a forecast range of 6 hours). The 5e7F (Fig. 1(b,e,h)) and 1s0dP (Fig. 1(c,f,i)) configurations both show a band of rain and cloud extending across the domain from southwest to northeast. Rainfall and outgoing fluxes of longwave and shortwave radiation are qualitatively similar in the two experiments and broadly reproduce the observed structures.

The similarities between the simulated storms disguise large differences in their microphysical structures. The vertical sections in Figure 2 show that inside the squall line 1s0dP has cloud-droplet numbers that are orders of magnitude smaller than those in 5e7F. The number concentrations of other hydrometeors also differ, for example 1s0dP has greater numbers of raindrops (Figs 2(c,d)). Figures 2(e,f) show that these differences are related to aerosol concentrations inside the squall line, where depletion during activation creates a low-aerosol environment. The longitude-time plot in Figure 2d shows that as the storm propagates a low-aerosol 'core' is maintained, despite the presence of more polluted air outside the squall line. The core region (where most rain falls) therefore has less aerosol and fewer cloud droplets than its surroundings. This "polluted-source – clean-core" structure cannot be replicated in the fixed-concentration simulations: in 5e7F, for example, there is more aerosol inside the squall line and production of rain therefore proceeds in the presence of higher concentrations cloud droplets.

The responses of cloud and rain to changes in aerosol are shown in Figure 3. The concentrations of cloud droplets in 5e7F are relatively large inside the squall line, compared to 1s0dP, whereas the concentrations of rain droplets are correspondingly smaller. There are analogous differences in the hydrometeor masses: the cloud-liquid water content is largest in 5e7F, and there is a corresponding deficit of rain water. Aerosol also affects the size of the hydrometeors (Figs 3(j-l)), with the mean-size of cloud droplets becoming larger as the aerosol concentration is reduced. The rain-drop sizes (the grey contours in Figs 3(j-l)) show the opposite trend, with the cleaner simulations associated with smaller rain drops (see also Fig. 4). This suggests that the rates of warm-rain processes differ between the experiments. In particular, because 5e7F has more cloud droplets it converts cloud to rain at a slower rate and therefore produces fewer rain drops. Conversely, 1s0dP has faster conversion of droplets which increases rain at the expense of cloud liquid. Despite the differences in composition between the two experiments, the effects on surface rainfall appear to be relatively small (Figs 3(a,d)) There is however a noticeable reduction in the heaviest rainfall rates (greater than 16 mm/h) in 1s0dP which we investigate in more detail below.

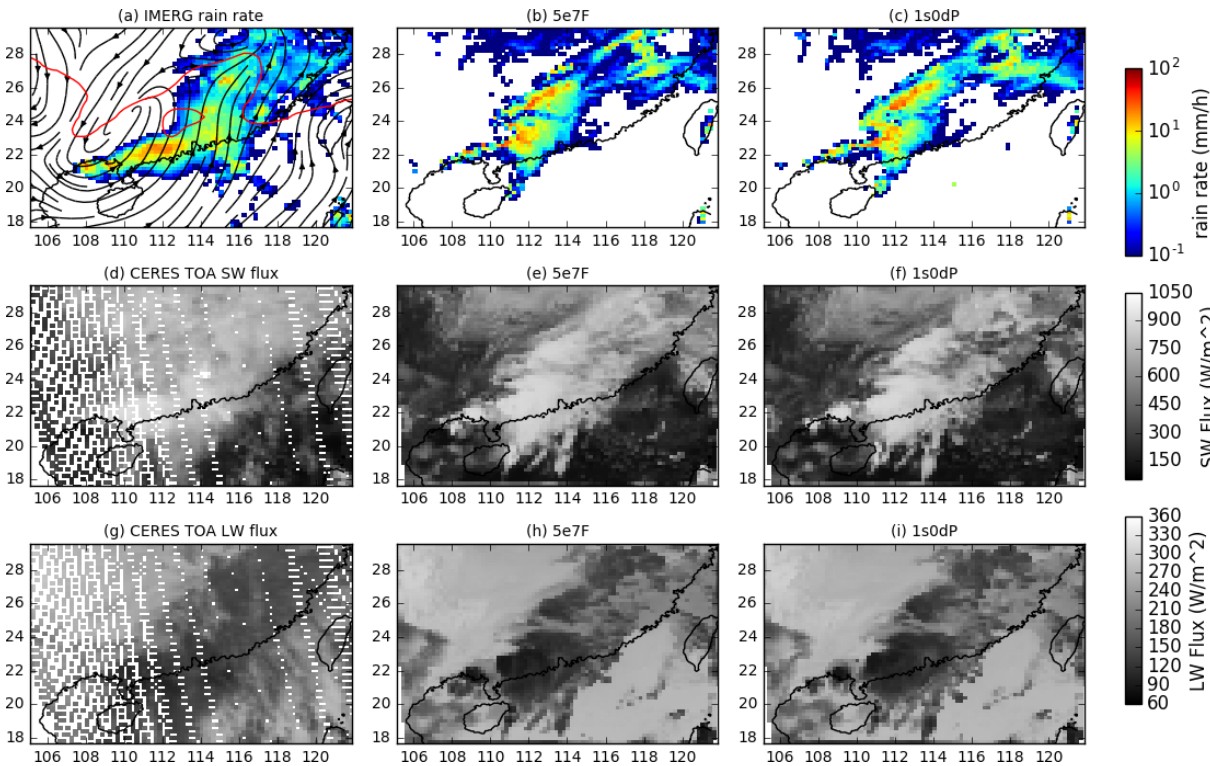

**Figure 1.** Surface rainfall rates (a-c), outgoing top-of-atmosphere shortwave (d-f) and longwave (g-i) in GPM IMERG (a), CERES-SSF (d,g) and simulations with the 5e7F (b,e,h) and 1s0dP (c,f,i) configurations. The black streamlines and red contour in (a) show the 850 hPa velocity field and the 315 K contour of equivalent potential temperature, respectively, in the ERA Interim reanalysis. The regional coastline is shown in black. All the fields shown are valid within at most 30 minutes of 06 UTC 20 May 2016.

By reducing the aerosol-number concentration in a fixed-aerosol experiment, we can assess whether one-way coupling can be 'tuned' to resemble the fully coupled simulation. Figures 3(g-h) show the evolution in a third experiment (5e6F) in which the aerosol has been reduced to a concentration ($5 \times 10^6$ m$^{-3}$) that is representative of the squall-line interior in 1s0dP. It can be seen that the reduced-number experiment bears closer resemblance to 1s0dP in terms of hydrometeor concentrations *inside* the squall line but, because it suppresses aerosol numbers throughout the domain, regions outside the squall line have cloud-droplet numbers that are lower than 1s0dP.

### 3.1.1 Sensitivities of hydrometeor-number concentrations

The mechanisms by which aerosol numbers affect cloud properties can be investigated by considering the evolution of cloud structure in a frame moving with the squall line. To this end, Figure 4 shows the profiles of hydrometeor concentrations averaged over a box around the centroid of surface rainfall. (The path of this box is shown by the black circles in Figs 3(a,d,g) and follows the region of heaviest rainfall.) The two low-aerosol experiments (1s0dP and 5e6F) have fewer cloud-droplets

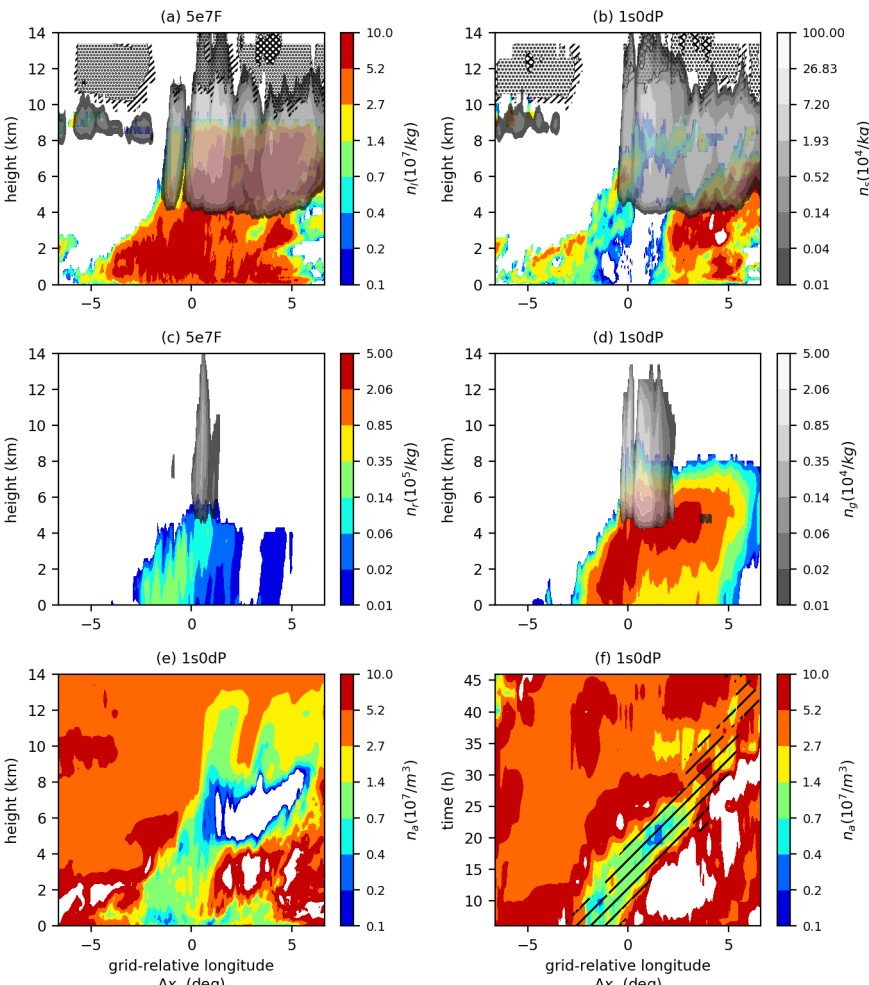

**Figure 2.** Simulated vertical sections, (a–d), and a (rotated-)longitude *versus* time plot, (e,f), of: hydrometeor number concentrations, (a,b); aerosol number concentration, (c,d). The vertical sections are along a line of fixed rotated-pole latitude (i.e.,the ordinate is relative longitude, $\Delta x_r$, in the model's coordinates). The coordinate is hybrid height. The longitude-time plot, (d), is at a hybrid height of 3405 m (approximately 3405 m above the local surface). Panel (a,c) shows the hydrometeor number concentrations for the 5e7F (fixed aerosol) experiment. Panels (b,d,d,e,f) are from the 1s0dP (aerosol processing) experiment. The colors show number concentrations, according to the adjacent scales: (a,b) cloud-droplets (colors), snow (grey), and ice (hatching); (c,d) rain (colors), graupel (grey); (e,f) aerosol. In (d), the hatched region indicates where the surface rain rate exceeds 0.1 mm/h.

and more rain-drops than 5e7F. The cloud-droplet number profiles in the 5e7F and 5e6F are relatively uniform below the homogeneous freezing level because aerosol number concentration is constant in these simulations. Differences in rain-drop numbers (orange) are largest close to the melting layer, between 4 and 5 km. The profiles indicate that most rain is produced at these heights, below which the profiles gradually taper towards the surface as drops evaporate. The cloud-water content (Figs

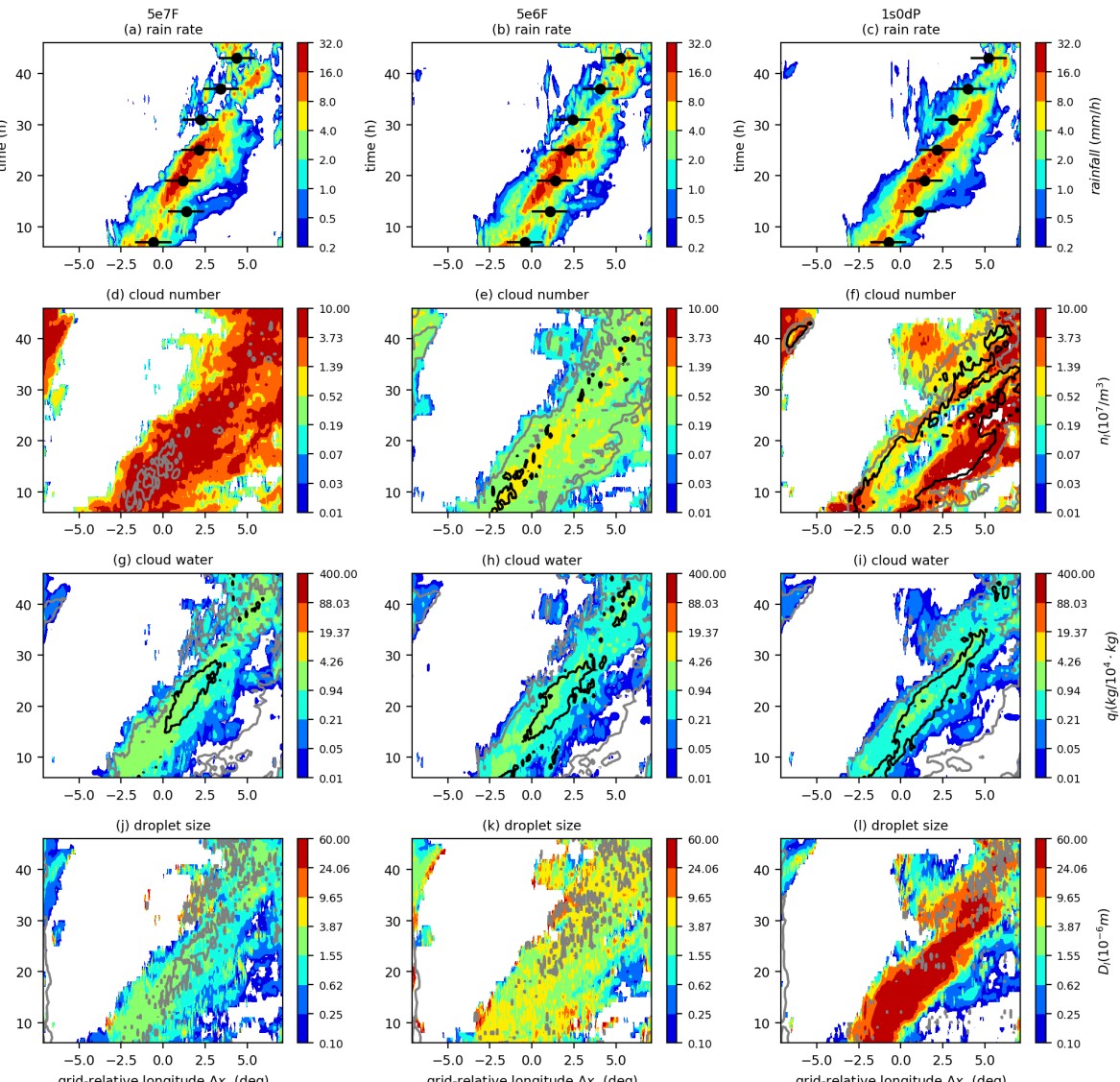

**Figure 3.** Rotated-longitude–time plots of: surface rainfall rate, (a,b,c), cloud-droplet number concentrations, (d,e,f), mass concentrations, (g,h,i), and cloud-droplet size (j,k,l) at a hybrid-height of 3405 m in the grid-relative longitude, $\Delta x_r$. The results from three models are shown: 5e7F (a,d,g,j); 5e6F (b,e,h,k); 1s0dP, (c,f,i,l); The colors show rainfall rate, the mass/number concentrations and the mean sizes, according to the adjacent scales. The solid black and grey contours are lines of constant rain-drop mass, number or size. The decorated black circles in (a,d,g) show the locations at three-hour intervals of a $2° \times 2°$ box centered on the moving centroid of the surface rainfall-rate field.

4(e-h)) also peaks in 4–5-km layer indicating that condensation of liquid cloud is most active at these heights. The dashed lines in Fig 4(a-d) show the profiles of rain-drop mean diameter. As noted above, the rain drops are smaller in the simulations

with fewer aerosol particles. The number of ice crystals (green) is smaller in simulations with fewer cloud droplets aloft. This is consistent with fewer cloud droplets leading to less nucleation of ice, via either homogeneous freezing or heterogeneous (immersion) freezing.

Because of the possibilities that ice and mixed-phase processes are affected by changes in cloud-droplet number, it is difficult to conclusively identify the primary mechanism dominating the increases in rain-drop number between the high- and low-aerosol experiments. We can attempt to disambiguate the mechanisms involved by considering three possible 'scenarios' for cloud-aerosol interaction, and the ability of each to explain the simulated changes in hydrometeor-number concentrations. The scenarios are as follows.

– *Warm-rain-process dominated* (thereafter, "Scenario-*W*" (*ScW*)): the number of cloud-droplets activated in 4–5-km layer is the dominant influence on the number of rain drops, via the droplet-number dependent autoconversion relation; more cloud-droplets causes a decrease in the rate at which cloud-droplets convert to rain drops, which leads to fewer rain drops.

– *Cloud-droplet freezing dominated (ScF)*: the number of ice crystals and snow aggregates present is the dominant factor;
more cloud-droplets leads to more ice particles (via increases in heterogeneous or homogeneous freezing), which leads to more rain drops due to melting snow.

– *Mixed-phased-feedback dominated (ScM)*: the rate of riming in the mixed-phase layer is the dominant factor; as the cloud-droplet number increases, the droplet size decreases, so riming of ice particles is suppressed; the number of graupel particles decreases, which leads to fewer rain drops due to melting graupel.

In addition to these three individual scenarios, combinations of the 'warm', 'cold' and 'mixed' mechanisms may be opperative in the simulations. For example, scenerios ScW and ScM could occur together, which would strengthen the effects of aerosols because both mechanisms suppress rain-drop number in less-clean environments.

If we consider each scenario in isolation, then the second scenario (ScF) can be ruled out as implausible (for this case) because Figs 4(a-c) show decreasing ice-crystal numbers (green) as aerosol decreases, but an increase in rain-drop numbers.
This is in contrast to the expected result of the freezing-dominated mechanism, whereby lower cloud-droplet numbers are associated with fewer rain drops.

The mixed-phased scenario, ScM, offers a possible explanation for the increased number of rain drops and is also consistent with simulated increases in graupel numbers as aerosol decreases. However, the vertical profiles in Figs 4(a-c) show that the increases in graupel between the high- and low-aerosol experiments are at least an order of magnitude smaller than the
increases in the number of rain drops. Hence, is it difficult to attribute the large increases in rain-drop number to the (much smaller) increases in graupel.

The purely warm-rain scenario, ScW, can also be used to explain the changes in rain-drop number. Moreover, it has the advantage that the droplet-number changes are sufficiently large to account for the increases in rain-drop number. (The very large 'reservoirs' of cloud droplets in all the simulation, mean that small fractional changes in the number of droplets, can

produce order of magnitude difference in the number of rain drops; particularly, if the strongly non-linear dependence of auto-conversion rate on droplet number is considered (Khairoutdinov and Kogan , 2000).) ScW also predicts the simulated tendencies for rain-drop size: faster warm-rain process can lead to more numerous and smaller-sized rain drops. We therefore

suggest that ScW is the scenario that offers the best interpretation of the simulation results.

This conclusion is also supported by the rain-drop number profiles in the 5e7F experiment (Fig 4a): in this case, autoconversion is suppressed and the rain-drop number at 4 km tends to a value that is consistent with the number of melting snow aggregates immediately above; the role of melting of frozen hydrometeors is therefore that they provide a lower bound on rain-drop *numbers* which is approached as the cloud-droplet number increases. Note that this does not imply that snow is

unimportant for the amount of rain. In fact, precipitating snow provides the *mass* flux into the melting layer from above. This is evident in the vertical profiles in Figs 4(e-h), which show that the rain-water content below the melting layer is limited by mass of snow immediately above. As the number of rain drops increases, the ratio of rain to snow increases because a larger mass of rain is needed to balance the snow-fall flux from above. In other words: in the cleaner simulations, the mass-flux from melting snow is transported by a larger number of (smaller) rain drops and a larger mass of rain resides in the column. *There-*

*fore, in ScW, warm-rain processes modulate the rain-drop number, and the rain-water content responds to this by increasing or decreasing so that the mass-flux of frozen precipitation from above is conserved.* This process is discussed in more detail in Section 3.1.3.

Further evidence supporting the relative importance of ScW is provided by an additional experiment, 5e6F_ACC, in which the aerosol-number concentration is $5 \times 10^6$/kg and warm-rain processes (auto-conversion and accreation) are turned off at

grid points where the temperature is warmer than the heterogeneous freezing temperature of rain ($-4°$C). In this simulation the purely warm-rain-mediated aerosol effects are completely suppressed, but the main mixed-phase processes (ice nucleation, freezing of rain, riming and snow–rain collisions) are still active and will be affected by aerosol changes in the same manner as in the full-microphysics simulations. Figure 4d shows that the rain-number profile in 5e6_ACC resembles 5e7F more closely than it does 5e6F. (Note that the cold-rain part of the profile *does* resemble 5e6F, because autoconversion and accreation are

permitted below $-4°$C.) Hence, if the aerosol is decreased but warm-rain process are not included, then no strong effects on rain-drop numbers are observed. Conversely, considering the differences between 5e6F and 5e6F_ACC: an effect similar to those of increasing aerosol can be obtained by turning off warm-rain processes. This suggests that the modulation of warm-rain processes is a crucial factor determining the rain-drop number response to aerosol. We note –however– that this does not imply the mixed-phase mechanism, ScM, is not important, but it is difficult to design a numerical experiment where melting of frozen

hydrometeors is suppressed without completely altering the microphysical structure of below the melting level. Hence, ScM could be acting in combination with ScW to determine the overall rain response.

### 3.1.2   Sensitivity of condensed-water paths

Because rain-drop fallspeed increases with drop size, the ScW mechanism has the following consequences for the effects of aerosols on the rain-water path and surface rainfall:

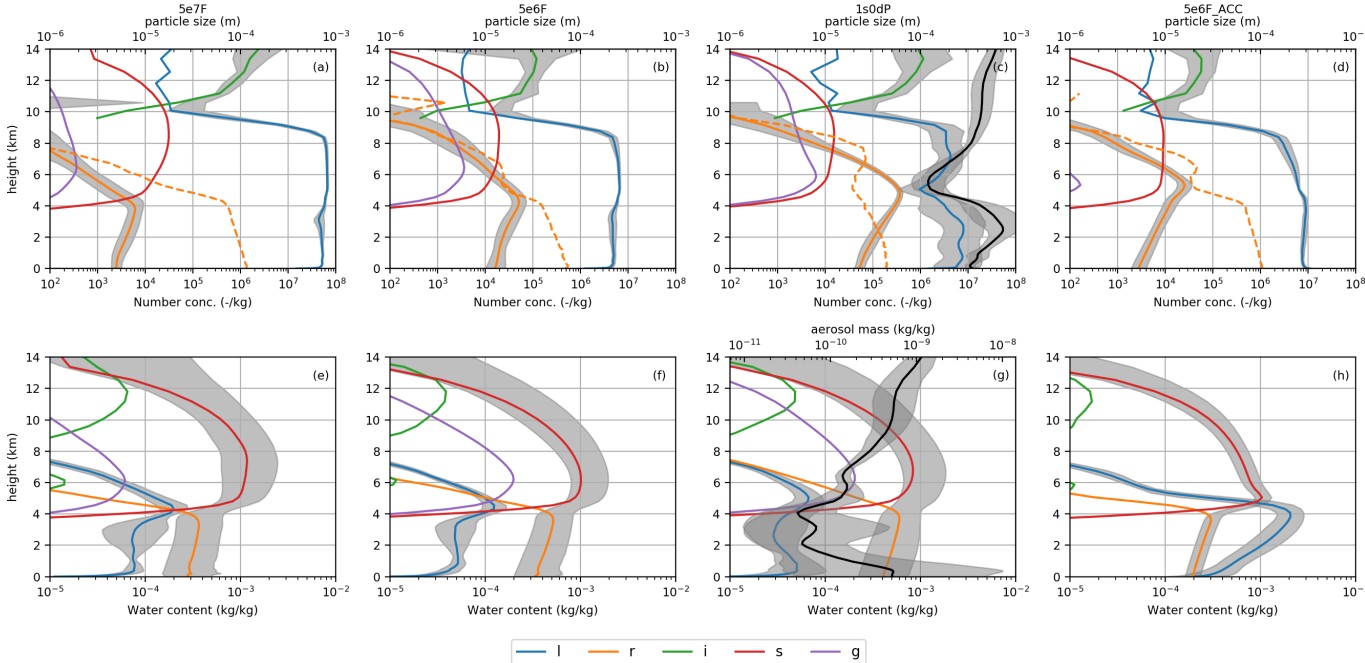

**Figure 4.** Vertical profiles of hydrometeor number concentrations, (a-d), and mass concentrations, (e-h), time-averaged within a $2 \times 2°$ box moving with the centroid of surface rain rate: (a,e) 5e7F; (b,f) 5e6F; (c,g) 1s0dP; (d,h) 5e6F_ACC. The colors correspond to the hydrometeor species according to the key shown on the left. The black lines in panels (c,g) show the profiles of the aerosol number and mass concentration in the aerosol-processing experiment (1s0dP). The grey regions show the variabilities (defined as $\pm 1$ geometric standard deviation) in the number and mass concentrations of cloud droplets, rain drops and the totality hydrometeor, according to whichever parameter has the largest variability at each point. In panel (c), the standard deviation around the aerosol number concentration profile is also shown.

- (E1) *for a given rain water path*, a 'clean' (low-aerosol) system will have smaller rainfall rate than a more polluted (high-aerosol) system;

- (E2) *for a given surface-rainfall rate*, a cleaner system will have a larger rain-water path than a higher-aerosol system;

in addition, rain-water path and cloud-water path are expected to vary inversely to one another at a fixed precipitation rate (since, if the rainfall rate does not change, any mass lost from cloud-droplets results in an increase in the mass of rain). The mechanism underpinning E1 is that smaller drops fall slower so if the rain water is held constant then the precipitation rate will decrease as the number of rain-drops increases. Similarly, E2 holds because if the rainfall rate is constant then a larger number concentration requires a larger rain-water path to provide this precipitation flux. A mathematical justification for E1 and E2 is given in Appendix A.

To evaluate whether the relationships E1 and E2 can be detected in the simulation output, Figure 5 shows the simulated condensed-water paths partitioned according to surface-rainfall rate. Firstly, note that Figs 5(a,b) show that, for fixed concentration of aerosol particles, higher rain water paths are associated with more cloud water and snow. We suggest that figuratively

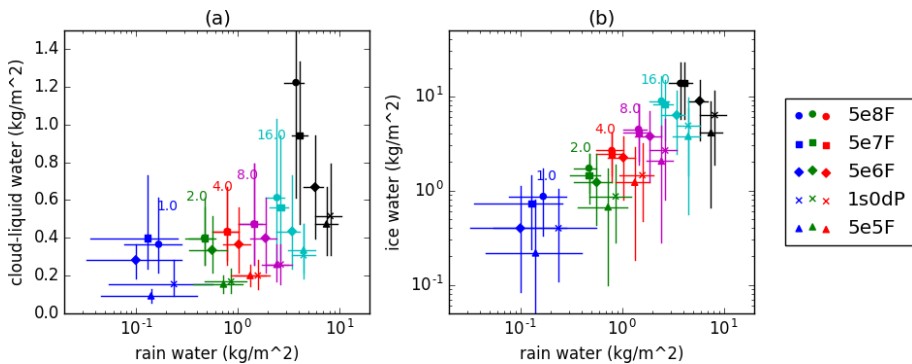

**Figure 5.** The scaling of (a) cloud-droplet liquid water path, (b) ice water path (snow and ice crystals), with rain water path, partitioned according to surface rainfall rate. The different symbols (circles, squares, etc.) denote different model experiments, according to the key shown on the right. The colors corresponded to logarithmically spaced surface rainfall intervals, the upper bound of each interval is indicated by the colored text on the plots (e.g, the red-colored points correspond to grid points where the rain rate was between 2 and 4 mm/h). The black points are for all rain rates greater that 16 mm/h. The horizontal and vertical bars show the inter-quartile ranges in each interval for the ordinate and coordinate variables.

speaking, these association can be considered as representing the internal 'microstructure' of the cloud system. For example, in this case, the structure of convective cores simultaneously results in high condensed water paths and high rainfall rates. Each experiment exhibits a different relationship between cloud-, rain-, and ice-water paths, and therefore generates clouds with
different microstructures. The differences in cloud microstructure are due to aerosol concentrations, particularly within the squall line. For a fixed rate of rainfall (color), lower-aerosol systems attain this rain rate with a less cloud water and more rain water and less ice water than higher-aerosol systems. Alternatively, by considering variations in rainfall along vertical lines in Figure 5a, we see that a relatively less clean system can support a higher rainfall rates for a given rain-water path. Both these findings are consistent with physically based expectations E1 and E2. This supports the conclusion that (for these simulations)
aerosols affect clouds by modifying the rate at which cloud droplets converts to rain.

### 3.1.3 The effects of ice- and mixed-phase processes on condensed-water paths

The possible effects of cold-cloud microphysics merit further discussion. Figure 5b shows that ice-water path varies inversely to rain-water path at fixed rainfall flux. Moreover, E1 and E2 require only that decreasing aerosol concentrations are associated with increasing numbers of rain drops. This can result from enhanced auto-conversion rate (as in ScW), but is also consistent
with an aerosol-induced increase in riming rates (ScM). (In the later case, fewer aerosols leads to larger cloud droplets, which rime more easily to create graupel, which increases melting.) The experiments here are not sufficient to demonstrate the extent to which this mechanism is contributing to the water-path differences. However, based on the order of magnitude differences between the number concentrations of rain and graupel, this mechanism is unlikely to be dominant. Moreover, a Supplementary Figure S1 shows that the 5e6F_ACC (no warm-rain processes) simulation does not reproduce the responses of rain- and ice-

water paths to aerosols seen in 5e6F. This at least indicates that if only cold-rain-mediated effects are permitted then the rain- and ice-water paths are not strongly affected by aerosol. Instead, it is more likely that the cold-cloud responses shown in Fig. 5b result from decreases in rain-liquid leading to more water being available for growth of ice and snow. Similarly the decrease in graupel with increasing aerosol seen in Figs 4(d-f) is consistent with reduced riming (due to decreased droplet size) and less heterogeneous freezing of rain. Further disambiguation of aerosol-effects, at the level of microphysical processes, is not possible with the experiments presented here.

## 3.2   Comparisions with observations

The preceding discussion demonstrates that the treatment of aerosols affects the microphysical structure of the squall line. However, the preliminary inspection of rainfall and long-wave fluxes in Figures 1 and 3 suggest that the hydrological and radiative impacts of these changes are not large (compared to forecast errors that are common to all the simulations). We now present a more detailed analysis of rainfall and top-of-atmosphere radiation which reveals that systematic differences do exist between the simulations. Figure 6 shows histograms of radar reflectivity, surface-rainfall rate and TOA SW-flux. The simulated histograms of reflectivity are increasingly shifted towards smaller values of dBZ as aerosol number decreases. The effects of these shifts are particularly pronounced in the large-dBZ tails of the distributions, where the cleaner experiments (1s0dP and 5e6F) have lower frequencies. This shift is related to a corresponding skewing of the rainfall-rate histograms towards lighter rain in the cleaner simulations. Reflectivity factor and rainfall flux are determined by moments of the drop-size distribution, hence their ratio contains information about the typical radii of the drops present. If the reflectivity is conditionally sampled based on the surface-rainfall rate, we can interpret the variability of reflectivity with aerosol number in each rainfall interval as being due to variability in the typical-drop size. The colored circles in Fig. 6b show the mean reflectivity in three rain-rate intervals. Because the clean-experiment histograms are skewed toward lower values of dBZ, the mean reflectivity in each rain-rate intervals decreases with decreasing aerosol number. This is consistent with lower cloud-droplet numbers giving rise to more numerous but smaller rain drops, because conversion of droplets to rain proceeds more rapidly. Cold-rain processes can also influence the occurrence of small reflectivity values and rain rates. For example, changes in the rate-of-production of graupel particles of different sizes could influence the distribution of drop sizes after melting, and therefore contribute to the skewing of the reflectivity histograms towards smaller values of dBZ.

The relationship between independent measurements of surface-rain rate and reflectivity is a directly observable way of quantifying the microphysical structure of the squall line. Moreover, in the simulations this relationship is strongly modulated by aerosol concentration. The simulated "(dB)Z-R" relationships (colored circles in Fig. 6b) can be compared to the relationship derived from the radar and IMERG measurements (black circles). Interestingly, despite biases in the underlying histograms of rainfall and reflectivity, the measurement-derived relationship is spanned by the simulations. These relations are structural properties that emerge from representations of microphysical processes in the models. Therefore, although the

simulated relationships may result from compensating biases[2], they still assess how well the models perform at capturing the observed co-variability of precipitation properties.

Measurements of top-of-atmosphere radiation can also be used to evaluate the performance of the simulations. Figure 6c evaluates histograms of outgoing-SW flux against the CERES-SSF measurements. The simulated distributions overestimate the frequency of occurrence of points with low reflected-solar flux (100–300 W m$^{-2}$) and underestimate the occurrence of fluxes greater than 600 W m$^{-2}$). The models show better agreement with the observations for fluxes that are intermediate between these two extremes (300–600 W m$^{-2}$). To understand which types of cloud cause the model biases in different parts of the histograms, the colored markers in Fig. 6c show the water paths of rain (squares) and cloud-liquid (circles), averaged over the three SW-flux intervals. The overestimated, low-flux peaks are associated with the presence of lightly or non-precipitating clouds with low ratios of rain-to-cloud, whereas the underestimated fluxes are in an interval that is dominated by columns with high liquid water paths. The high liquid-water content regions are due to the passage of the squall line across the domain. This suggests that the overestimates at low-fluxes are due to insufficient stratiform cloud cover in the regions away from the deep-frontal clouds. This error is least in the most polluted experiment (5e8F), which is consistent with the suppressed autoconversion in the experiment causing an increase in the prevalence or longevity of liquid water clouds in that simulation. From the short-wave flux histograms, it is evident that 5e8F redistributes the low-rain-water content pixels into brighter parts of the SW distribution, where they coincide in terms of reflected flux with the deep-frontal clouds. Conversely, the underestimates of shallow-cloud amount are largest for the cleanest of the fixed-number experiments, because rapid conversion of cloud to rain reduces the amount of liquid cloud these simulations can produce. It is noteworthy that the most polluted experiment (5e8F) shows the best agreement with the observations for TOA-SW flux but has relatively poor performance for radar reflectivity and surface rainfall rate. This may be because larger cloud-droplet numbers suppress drizzle, which is beneficial for stratiform clouds, and increase ice-crystal number –which is beneficial for the SW reflected from cirrus anvils– but simultaneously increase heavy-rain production in the convective-core regions[3].

To investigate further the cloud changes that are responsible for the differences in TOA radiation between the experiments, Fig. 7 shows the total condensed water path in each of three SW-flux intervals for the MODIS/CERES observations and a subset of the simulations. The fields are compared for a single overpass time of the satellite. For each SW-flux interval, grid-points with radiances outside that interval have been masked. The most polluted simulation, 5e8F, produces the most realistic field of clouds. In particular, it is the only configuration that is able to produce a region of realistically bright stratus in the wake of the squall line (in the north-west corner of the domain). Stratus clouds also exists in the other simulations (eg, Figs 7(e,h,k)), but are less bright than in the retrievals.

The regions with deep ice clouds (which for the models can be defined as where the ice water path exceeds 20 percent of the total water path[4]) are shown by the black lines in Figs 7(c,f,i,l) As the aerosol-number concentration decreases, the number of grid-points with ice cloud which have radiances greater than 600 W/m$^2$ decreases (Figs 7(f,i,l)). The SW-fluxes

---

[2]In this respect, if it interesting to note that the model with the best rain-rate histogram (5e5F) has the largest discrepancy from the observed Z-R relationship.

[3]This is evidence of a structural error in the model, which we are grateful to an anonymous reviewer for bringing to our attention.

[4]For the MODIS retrievals we can demarcate an ice region as those pixels for which MODIS retrieves either mixed-phase of ice cloud (Fig. 7c).

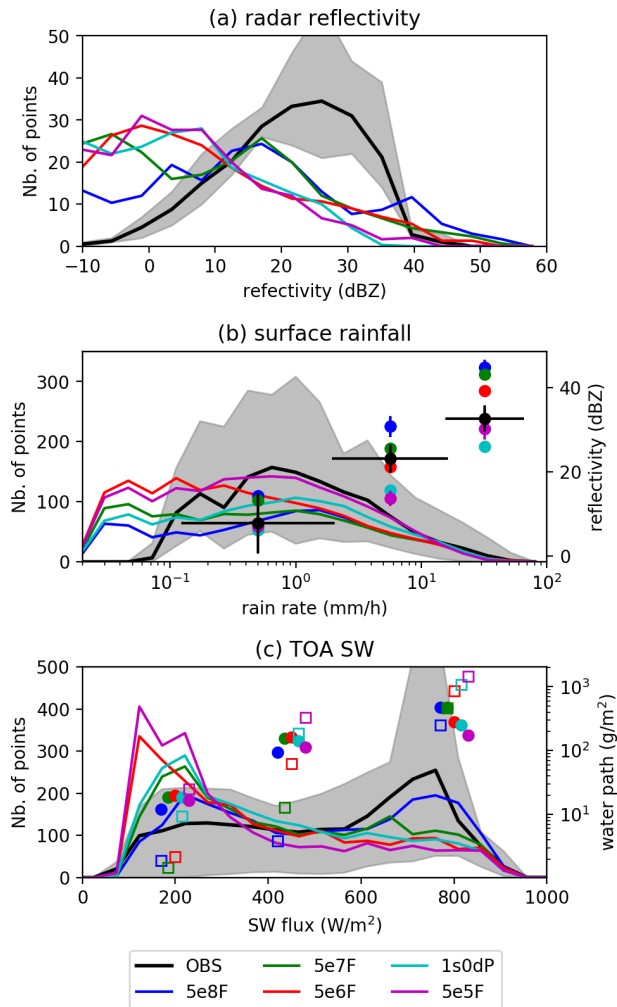

**Figure 6.** Time-mean histograms (solid lines) of: (a) radar reflectivity, (b) surface rainfall rate, (c) outgoing SW-flux, in the model experiments compared to the observations. The grey shaded area shows the range (from minimum to maximum number of counts) for the observations, in each variable-bin. In panel (b) the colored circles show the mean radar reflectivity in three surface-rainfall intervals for the simulations (colors) and IMERG retrievals (black). The horizontal bars on the IMERG-points show the lengths of each intervals. In panel (c) the colored symbols show the mean cloud-water paths (circles) and rain-water paths (squares) in three contiguous SW-flux intervals, for the simulations.

reflected from these points fall instead into the low-radiance intervals, which are overpopulated with pixels compared to the MODIS/CERES estimates. Figures 7(j-l) show the results from the processing experiment, 1s0dP. The water paths in this experiment are intermediate between the fixed-number experiments: behind the squall line, it resembles the cleaner fixed-number experiment (5e7F); within the squall, it has fewer high-condensed-water path (higher brightness) columns than 5e7F.

The differences between the rainfall frequency distributions shown in figure 6b suggest that there may be detectable effects of aerosols on the mean properties of surface rainfall during this case. Since it such averages, rather than the statistical distribution

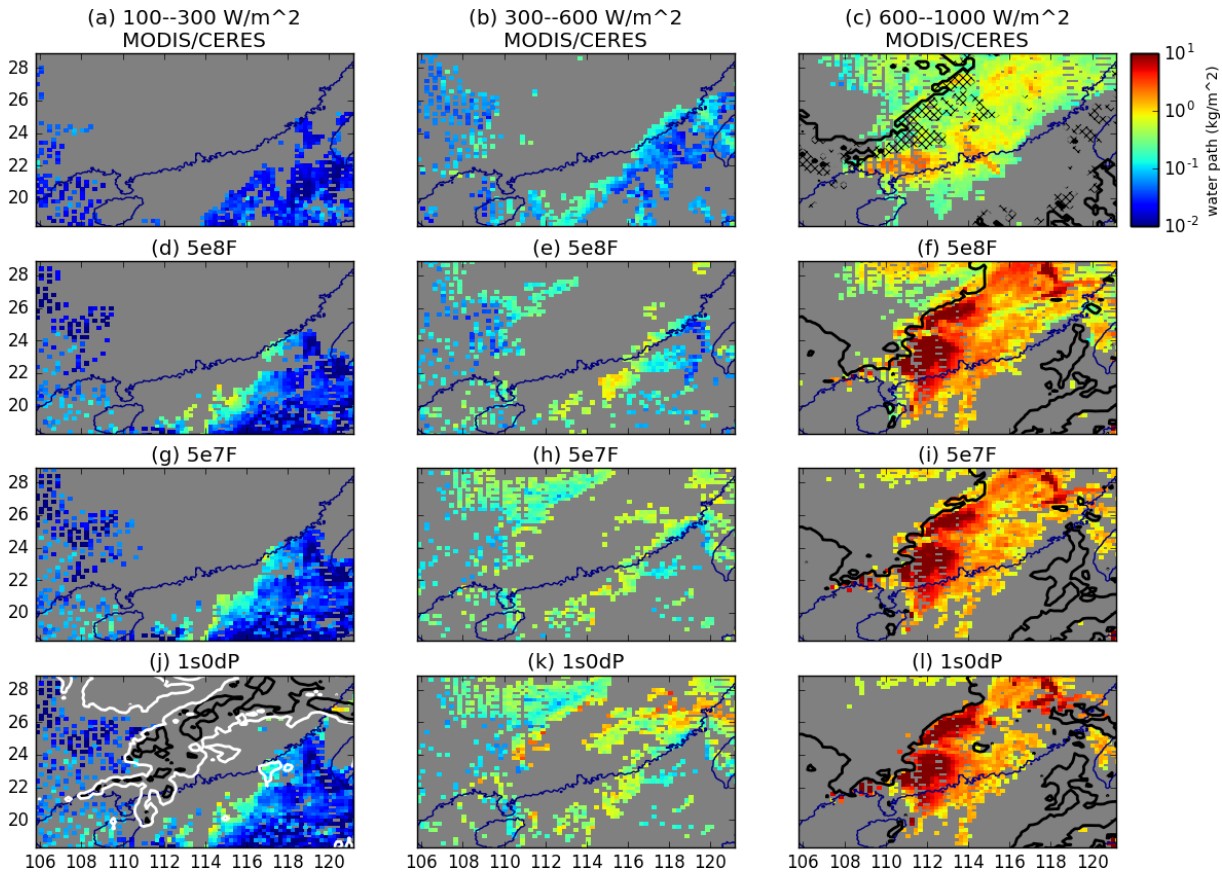

**Figure 7.** Total condensed water path (all species of hydrometeor) in the MODIS observations (a-c), 5e8F (d-f), 5e7F (g-i) and 1s0dP (j-l). The fields are partitioned into three SW-flux intervals: fluxes between 100 and 300 W m$^{-2}$ (a,d,g,j); fluxes between 300 and 600 W m$^{-2}$ (b,d,h,k); fluxes greater than 600 W m$^2$ (c,f,i,l). For each panel, points outside the corresponding interval of fluxes have been masked out in grey. The black contour in (c,f,i,l) encloses the regions where the fraction (by mass) of the total condensed water path that is ice exceeds 20 percent. In panel (j) the contours show where the column integrated aerosol in the lowest 5 km of the atmosphere are 20 percent (black) and 80 percent (white) of the domain-mean aerosol path.

of rainfall which are of primary importance of regional-scale hydrology we conclude this section by examining the time series of average rainfall characteristics in Figure 8. To characterise rainfall, we adopt the commonly used approach of decomposing the hourly instantaneous surface-rainfall rates into contributions from the areal coverage of rainfall (rainfall frequency) and the mean intensity of rainfall at rainy points (rainfall intensity). These quantities are defined relative to a threshold minimum rain rate of 0.1 mm/h (chosen to reflect the lower limit of the IMERG retrievals). Hence, if $A$ is the total area of the domain, then the frequency, $f$, is given by $f = \sum^{>} \Delta_{\mathbf{x}}/A$, where the summation is over points $\mathbf{x}$ where rainfall exceeds the threshold, and $\Delta_{\mathbf{x}}$ is grid-box area. Similar, the intensity $I = \sum^{>} p_{\mathbf{x}}\Delta_{\mathbf{x}}/Af$, where $p$ is the surface-rainfall rate, and amount $P = I \times f$.

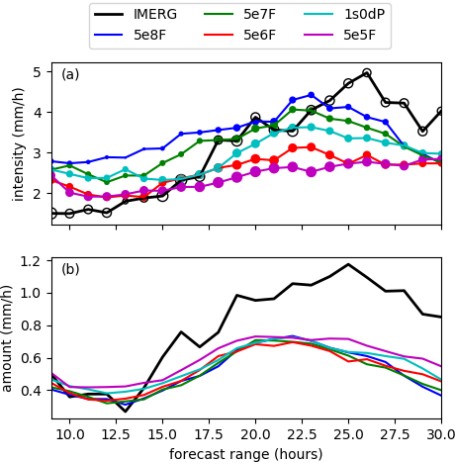

**Figure 8.** Time series of (a) domain averaged rainfall intensity (lines) and frequency (circles), (b) rainfall amount, in the models (colors) and IMERG retrievals (black). The time is given in hours elapsed since the initialisation of the forecasts (00 UTC 20 May).

Figure 8a shows that rainfall intensities (solid lines) increase with increasing aerosol concentration. The most polluted simulations have surface rainfall that is up to 50 percent more intense than the least polluted experiments. The aerosol-processing experiment has intensities that are intermediate between the least-polluted and most-polluted experiments. This is consistent with the observation made above that for lower rain rates the histogram of rain rates for 1s0dP is similar to the higher-fixed-number experiments (5e7F and 5e8F) whereas for higher rain rates it is similar to the least polluted scheme (5e5F).

The sizes of the circles in Fig. 8 show the relative differences in the frequency of rainfall for each simulation. It can be
seen that the cleaner experiments have rainfall over a larger fraction of domain, which is consistent with the rainfall (and reflectivity) histograms becoming progressively more shifted towards lower rain rates (and reflectivity factors) as the aerosol number concentration decreases. The overall dependence of surface rainfall on aerosol can therefore be characterised as a transition towards a regime of more-frequent, less-intense rainfall as the aerosol number increases. By capturing both high- and low-aerosol concentrations in the same domain, the aerosol-processing experiment is able to combine aspects of both the low-
and high-intensity regimes and therefore gives surface rainfall properties that are intermediate between the clean and polluted extremes. The opposing tendencies in $f$ and $I$ are consistent with Miltenberger et al (2018) who found that convective-cell size increased (indicating more intense rainfall) whereas the number of cells decreased with increasing amounts of aerosols (indicating a smaller in the rainy area). Finally, in Fig. 8b, we see that the amount of surface rainfall is *marginally* higher in cleaner simulations, because (for this case) rainfall amount is slightly dominated by the rain-frequency differences, offset
by an opposing tendency due to the decrease in intensity with aerosol number. It should also be noted that, because of the opposing influences of aerosols on $f$ and $I$ the difference in rainfall amount between the least- and most-polluted simulations are relatively small (of the order of at most 20 percent).

## 4 Conclusions

We have investigated how the representation of cloud-aerosol interactions influence simulations of a heavy-rainfall event over south China. Experiments with fixed-aerosol numbers, which spanned a range of ambient-air conditions (from relatively polluted to extremely clean), were used to demonstrate the effects of aerosols in one- and two-way coupled modeling frameworks. In the one-way coupled experiments the aerosol populations are not modified by cloud processes. In the two-way coupled model, the interstitial aerosols are depleted by activation of droplets and re-populated when hydrometeors evaporate. Satellite retrievals and ground-based radar measurements have been used to place the inter-model spread (and hence uncertainties in cloud-aerosol effects) into the context of measurable properties of the evolving system of clouds. The comparisons to observations show that forecasts with lower aerosol concentration give better predictions of histograms of hourly instantaneous rainfall rates. However, the same configurations underestimate the observed fluxes of short-wave radiation in regions where rain-water paths are large, and also underestimate reflected SW fluxes in areas with lightly precipitating clouds and low liquid-water paths.

We have shown that, for the simulations performed here, the dominant mechanism by which areosols influence clouds and precipitation is a 'warm-rain' pathway, whereby reducing aerosols decreases the timescale for producing rain and increases rain water at the expense of cloud droplets. This effect is particularly evident if the water paths of cloud and rain are compared at approximately constant values of the rainfall rate (because cloud-water path and rain-water path vary inversely to each other in a given rain-rate interval). Conversely, the same analysis shows that the lower the aerosol loading, the greater the rainfall that can result for a fixed cloud-water path. The simulations performed do not place an unambiguous constraint on aerosol effects mediated by mixed-phase processes (for example, the affects of aerosols on riming), but experiments with direct cloud-to-rain conversions turned off suggest that warm-rain processes are at least essential for the simulated cloud responses. For the regime (organised warm-sector convection, with large-scale forcing) and model considered, melting of ice crystals provides a lower bound on achievable rain-drop numbers, and an upper bound on the achievable rain-water paths.

Reductions in the mass of water suspended as cloud-droplets are accompanied by decreases in the brightness of the simulated clouds, particularly in regions away from any deep ice-clouds. Aerosols also alter the properties of rainfall reaching the surface: in the more polluted simulations, the rainfall is more intense (with higher rainfall rates when rain occurs) but occurs over a smaller spatial area. As a consequence of these competing changes, the amount of rainfall reaching the surface (the domain mean) is relatively insensitive to aerosol concentration. This latter point may have been anticipated because on the timescales of individual weather events the amount of rainfall is strongly constrained by large-scale convergence into the rainy regions at low levels.

The changes in rainfall intensity and rain fraction are manifestations of underlying changes in the probability distributions of surface rainfall. We have shown that simulations with fewer aerosols have more frequent lower rain rates than polluted experiments. A corresponding reduction in the occurrence of high values of radar reflectivity factor with decreasing aerosol number was detected when the simulated radar reflectivity factors were compared to measurements

The inclusion of two-way coupling between aerosols and clouds was shown to qualitatively change the simulations, in a manner that can not be replicated in a fixed-number experiment. In terms of the quantities assessed, the processing experiment

is intermediate between a relatively polluted fixed-aerosol experiment (with the same initial aerosol concentration) and a much cleaner simulation with a lower number concentration. This happens because activation removes aerosols from the ambient air, so the presence of deep-convection maintains a low-aerosol 'core', within which most the heavy rain forms. Rainfall is therefore produced mainly from cloud droplets which are less numerous than would be the case if the aerosol number was unaffected by activation. The result is that in a two-way coupled experiment the 'warm-rain' pathway to more vigorous then it would be in fixed-aerosol number experiment with the same initial conditions. The aerosol processing experiment therefore resembles a 'cleaner' fixed-number simulation in terms of the partitioning of water between cloud and rain inside the squall line. Outside the squall line, the opposite situation prevails: the aerosol population recovers towards the environmental value so any shallow clouds are in relatively polluted environments compared to squall line's interior. Hence, stratiform precipitation is produced from clouds with high cloud droplet numbers. The ability to represent different cloud-droplet numbers in different parts of the domain (in a way that depends on precursory cloud processes) may well be the main advantage of aerosol-processing models. Here we have shown that capturing these dependencies has novel consequences for simulating the structure of organised convection and affects both the hydrological and radiative impacts of such systems, in a manner consistent with –but not fully replicable by– one-way coupled simulations with tunned aerosol concentrations. However, despite the changes in the microphysical structure of the squall line, e.g., the occurrence of a 'clean-core', the overall impact of coupling on model performance against the metrics considered is small, particularly when compared to the overall biases present in all the model configurations. Hence the usefulness of the additional complexity of two-way coupling for weather forecasting is still moot, and will benefit from analysis of further cases in future work.

Two aspects not addressed in this paper may form the basis for future work: firstly, our analysis has been limited to a single case; secondly, we have omitted any discussion of ice-nucleating aerosols. Although single cases are useful for identifying mechanisms, they need to be supplemented by extended trails to quantify the affects on model performance. As part of such trails, it will be necessary to use observed (or at least re-analysed) aerosol concentrations to drive the models with realistic cloud nuclei. Moreover, in this paper, we have not considered the effects of ice-nucleation parameterizations –we plan to address this in the future as part of forecast trails with a range of microphysical configurations.

*Code and data availability.* The CERES/Aqua Level 2 Single-Scanner Footprint Edition 3A observed TOA Fluxes can be obtained from the Atmospheric Sciences Data Center at NASA Langley Research Center: https://ceres-tool.larc.nasa.gov/ord-tool. The MODIS/Aqua Collection 6 Level 2 Cloud Product data can be obtained from the Level-1 and Atmosphere Archive and Distribution System Distributed Active Archive Center in the Goddard Space Flight Center: https://ladsweb.modaps.eosdis.nasa.gov/archive. The reflectivity measurements from the Guangzhuo radar and the postprocessed model data can be obtained from the SCMREX data archive: http://exps.camscma.cn/scmrex The Integrated MultisatellitE Retrievals for GPM (IMERG) can be obtained from NASA's Precipitation Processing Center: ftp://arthurhou.pps.eosdis.nasa.gov//gpmdata. The Python code used is available for download from https://code.metoffice.gov.uk/trac/home.

 **Appendix A: The relationships between rain-water path, aerosol-number concentration and surface-rainfall rates**

The physical reasoning underpinning both E1 and E2 is that the rainfall flux, $P_r(z)$, at a height $z$ above the surface, is given related to the number concentration of rain drops, $n_r$, and the rain-water content, $\rho_r$, by $P_r(z) \sim n_r^{-\alpha} q_r^{\beta}$. The parameters $\alpha, \beta$ are positive constants which depend on the shape of the drop-size distributions, and the space- and time-scale used to define the rainfall rate (derivations can be found in Furtado et al (2015)). In approximately steady-state situations, as can be assumed to hold inside regions of persistent vertical motion, conservation of mass implies that the precipitation flux is approximately independent of height. Hence, the vertical profiles of $n_r(z)$ and $\rho_r(z)$, above a given point on the ground, are constrained by the rainfall rate at the surface below: $P_S \sim n_r^{-\alpha}(z) q_r^{\beta}(z)$. For both the ScW and ScM scenarios, a reasonable assumption is that $n_r$ scales with $1/n_a$, where $n_a$ is the number of aerosols. Hence, $P_s \sim n_a(z)^{\gamma} q_r^{\beta}(z)$, for $\gamma > 0$, from which both E1 and E2 follow by holding one of either $q_r$ or $P_S$ to be constant.

*Author contributions.* KF conducted the model experiments and analysed the results. YL and TJ conceived and instigated the work. YL provided SCMREX data from the Guangzhou Radar. PF provided scientific oversight throughout. AH and others developed the microphysics scheme code. All authors worked on the manuscript text.

*Competing interests.* None known.

*Acknowledgements.* This work was funded by National (Key) Basic Research and Development Program of China (2018YFC1507400). Kalli Furtado was supported by the UK-China Research and Innovation Partnership Fund through the Met Office Climate Science for Service Partnership (CSSP) China as part of the Newton Fund.

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
