# Peer review of "The effects of cloud-aerosol-interaction complexity on simulations of presummer rainfall over southern China"

_Atmospheric Chemistry and Physics, 2019_

## Referee Comment (RC1) · Anonymous Referee #3 · 1 Oct 2019

1). general comments The paper entitled "The effects of cloud-aerosol-interaction complexity on simulations of presummer rainfall over southern China". Paper analyzed the model simulations separately using the one-way fixed aerosol and two-way coupling aerosol system to evaluate the effects of cloud-aerosol interaction complexity on one heavy rainfall event. Also, the model simulations were validated with the radar and satellite observations. This study fellow the work of Miltenberger et al. (2018) to focus on one different precipitating system over China and might contribute to the fields. But the analyzing results in the paper left some to be improved (please see specific comments) so I would recommend a minor revision before it can be considered to accept for publication.
2). specific comments 1. Page 2-3, please illustrate the new thing/work of this study compared to the work of Miltenberger et al (2018) since both you are using the same microphysics scheme with the two-way aerosol coupled system to evaluate the effects of cloud-aerosol interaction.

2. Page 4 Line 42-44, the greater/lower number concentration is not easy to tell from Figure 2(a),(b) attributed to unclear and unlabeled colored contours;

3. Page 6 Figure 3, please add the plots of cloud droplets size in terms of three model runs for further comparisons. Also, please list another label (upper) to the color maps of panels (a),(c),(d),(f),(g),(i);

4. Page 6 Line 23-25, "the amount of rain ..... by the minimum cloud-droplet concentrations .....". Please check whether cloud concentration or size is the dominant parameter for the parameterization of auto-conversion from cloud to rain in CASIM microphysics scheme;

5. Page 6 Line 27-29, please add the initial vertical profile of aerosol concentration for checking. Perhaps the depletion of aerosol near melting layer was resulted from great droplet activation over there;

6. Page 7 Figure 4, please correct the labels of panel(d),(e),(f) and add the aerosol mass vertical profile in panel (f). Also, please check why the vertical profile of cloud number concentration was ceased between 105 and 106 per kg.

7. Page 7 Line 14, please explain the factor of 1/8.

8. Page 7 Line 55-58, please prove that lower cloud droplet number giving rise to smaller raindrops. Actually, lower CCN number could result in larger cloud droplets and higher collision efficiency, and then enhance auto-conversion and accretion processes together with more warm rain. Thus, the shifts of reflectivity factor and rainfall-rate toward smaller values in the cleaner model runs should be related to less cold rain production (like less cloud water, reduced freezing, and riming rates). Please modify

ACPD
the sentences.

9. Page 7 Figure 6(b), please check the lower limit of surface rain rate for model simulation was identical to the observation ( $\sim$ 0.1 mm/h).

10. Page 7 Figure 6, please explain that the most polluted simulation (5e8F) produced the good agreement in the TOA SW against the observation, but obtained the poor results over reflectivity factor and surface rain rate. Also, please discuss the role of 1s0dp simulation, is any difference between the one- and two-way aerosol coupling model run?

11. Since the title/motivation (Page 3 Line 8-16) of this study was to evaluate the effects of cloud-aerosol-interaction complexity, some further comparisons to depict the different impact between one- and two-way coupling aerosol systems are required. What is the benefit to adopt the two-way aerosol coupling on model simulations?

12. Pages 10-11, please give suggestions to the future operational weather forecast models if adopted the fully coupled cloud-aerosol-interacting system as illustrated in Page 3 Line 8-16.

3). typing errors 1. Page 2 Line 44-45, grammar typo; 2. Page 12 Line 63, 78-80, please complete the information of cited papers.

ACPD

---

## Referee Comment (RC2) · Anonymous Referee #1 · 5 Oct 2019

This manuscript presents numerical simulations of a heavy-rainfall event over south China with a focus on how the configuration of initial/background aerosol conditions influence the development of clouds and precipitation. Two sets of numerical experiments were conducted: One with the aerosol particle number concentrations kept constant, and another with full aerosol-cloud interactions. For the later case, the aerosol particles are allowed to complete due to nucleation and regeneration after evaporation of hydrometer. The study is interesting and could be an important contribution to understand how deep convection is affected by anthropogenic aerosols. However, I think that the experiments designed in the present manuscript are unable to show unambiguously the mechanisms behind different response of cloud and precipitation

to different configuration of initial/background aerosol. This is because, in the case with fully coupled aerosol-cloud interaction, the background aerosol concentration was variable, not a constant as in other tests. So, I suggest it might be wiser to add another test with the concentration of aerosol varying but without recycling after evaporation of hydrometeor. In addition, the manuscript is also lack of necessary explanations of the physical mechanisms induced by different aerosol loading. Therefore, I think the article needs a major revision before it can be considered publishable in ACP. More specific comments are as follows.

Specific comments:

1. Page 1, line 4-6: “simulations with aerosol concentrations held constant are compared with a fully coupled cloud-aerosol interacting system to isolate the effects of processing on a line of organised-deep convection”: It is difficult to draw conclusions about differences between such experiments, because not only the aerosol particle number concentrations are different, in the case with processed aerosol, coarse mode particles are added due to cloud processing, resulting in more effective giant CCN. To investigate how aerosol particles processed by clouds influence the development of future clouds and precipitation, I suggest to run the model with the coupled cloud-aerosol system but without recycling of aerosol back to the clouds.

2. Page 7: the caption of Fig. 2 should be in more concise form;

3. Page 8: in the first line of caption Fig.3, Also in some other places, “hydrometeor” should be cloud droplets. Hydrometeor also includes ice phase particles. At the top of figure, “cloud number” should be cloud droplets number;

4. Page 9: line 7-8: “The rain maxima are coincident with peaks in cloud-water 15 content (Figs 4(d-f)), indicating that condensation of liquid cloud is also most active in the 4–5-km layer”: what does this mean? where is (d)?

5. Page 9: line 8-10: from Fig. 4, it can be seen that rain mass is to a large extent

depends on snow, so the statement in these lines could be misleading;

6. Page 9: line 11-12: what do you want to tell?

7. Page 10: Fig. 4: why in (a) -(c) the number concentrations of cloud droplets are almost constant from about 8 km to the ground?

8. The last 2 lines in Page 10: "…shown in Figure 5a support the conclusion that (for this case) aerosol concentration affects the simulated clouds by modifying the rate at which liquid cloud converts to rain drops": in mixed-phase convective clouds, the conversion of cloud droplets to rain is not straightforward, it needs a detailed analysis, not just a guess.

9. Page 10: Fig. 5: what are the black symbols representing for?

10. Page12: Fig. 6: what are the parameters of the left axis?

11. Page 15: line 9-10: This needs a detailed analysis.

Technical corrections:

1. Page 3, line 34: change "are" to "and";

2. Page 8, line 9: "F5e6" should be "5e6F"?

3. Page 10: in the caption of Fig. 4, "d-f" should be "e-g"; "2x2°" should be "2°x 2°";

4. Page 15: Fig. 8: The unit of rain amount is not mm/h.

---

## Referee Comment (RC3) · Anonymous Referee #4 · 6 Oct 2019

Review of "The effects of cloud-aerosol-interaction complexity on simulations of presummer rainfall over Southern China" by Furtado et al.

This manuscript investigates aerosol-cloud interaction in a squall line case in Southern China. Two types of simulations are performed: one is to use prescribed aerosol concentrations, and the other is to include aerosol recycling after cloud-processing. The purpose is to study if the cloud-processed aerosols (as there is a transition from accumulation mode aerosols to coarse mode aerosols after the collision-coalescence process) can change the precipitation of the squall line case. It is concluded that aerosol recycling after cloud-processing can affect the precipitation through the enhancement

of warm rain process.

I think it is very interesting to show that the cloud-processed aerosols can promote the warm rain process. It is also interesting to see the "polluted source - clean core" structure of the system. However, I think the manuscript needs major revision before it can be published in ACP.

Major changes:

1. A more detailed and logical discussion on the microphysical processes should be provided.

The manuscript concluded that the cloud-processed aerosols can change rain formation mainly through the warm rain process, not through the melting of ice phase particles. However, the fact that "differences in rain-drop numbers are largest close to the melting layer (page 9, line 5)" is an indication that changes in rain in the simulation with cloud-processed aerosol is somehow related to the melting of ice phase particles. The authors did point out in a later paragraph (page 9, line 20-22) that, if the melting of ice phase particles is dominant, then more rainfall should be associated with high cloud droplet number and increased ice particle production. This study did not find these characteristics, so it is concluded that the melting of ice phase particles is not important. I think the manuscript should be revised to explain these microphysical processes in a more detailed and organized way. The conclusion that the cloud-processed aerosols can change the precipitation through warm rain process instead of the melting of ice phase needs to be explained in a very clear way.

I can see from Figure 5b that rain rate is related to ice water path. But this study seems to think that the ice phase process has secondary importance.

The ice phase processes need more explanations. For example: page 6, line 14, 1s0dp has more rain, but less snow. Why? The manuscript should at least provide a short explanation why snow particles has lower concentration in this case. Note that all the

explanations should be consistent with the results shown in Figure 4.

Some of the discussions related to ice phase processes might be wrong. For example: page 9, line 16-17, there is more graupel in the low aerosol experiment (because more rain is available for freezing). I would think the opposite: there is more graupel in the low aerosol experiment, so there is more rain from the melting of graupel. I strongly suggest the authors explain the mechanisms in a logical and clear way so that we can understand both the warm and cold cloud processes. Page 9, line 14-15, it is hard to understand this sentence.

Page 9, many characteristics of rain is discussed, including rain drop number concentration, the amount of rain, rate of rainfall, (line 29), and rain water (line 30). I suggest the authors explain these microphysical characteristics step by step. For example, I can easily understand that the cloud processing of aerosols can lead to changes in rain number concentration. But how the rain water and rain rate are affected should at least be explained.

Page 9, line 12: the "minimum" cloud droplet concentration. What does this minimum mean? Processes that can reduce cloud droplet concentration include collision-coalescence, entrainment and evaporation. Why would this minimum be so important? 1s0dp has cloud-droplet numbers that are orders of magnitude smaller than those in 5e7F. I think this is really a significant effect due to the cloud-processed aerosols. The manuscript should emphasize this point.

2. The setup of the model is not described very well.

Firstly, is the immersion freezing process affected by the different setup of aerosols? For example, it is said on page 4, line 26: a fraction of droplets are specified as ice. So for the 5 simulations in this study, cloud droplets have quite different number concentrations, then do they have quite different ice concentrations? Secondly, what is the use of the 5e5F and 5e8F simulations in this study? 5e5F has a really low aerosol concentration, which is probably unreal. In figure 8, if the 5e5F case is removed, then

the other simulations do not show much difference in precipitation amount. 5e8F is actually not quite "polluted".

3. Some of the main findings are not reflected in the abstract.

Line 7: what kind of vertical structure? What kind of statistics of surface rainfall? I think the writing could be more specific. For example, a more frequent and less intense rainfall are associated with higher aerosol concentration. This kind of clear results should be mentioned in the abstract. In addition, "a modulation by aerosol of the time scale", what kind of modulation? Are the evaluations of the simulations with observation data good? These should also be mentioned in the abstract.

Minor changes:

Figures 1-3 are too small. Especially the words and numbers in these figures are too small.

Figure 7, if the 20% criteria are changed, say, to 25%, are the results still robust?

Figure 8, (a) and (b) are missing in Figure 8.

Page 9ïijŇline 14, "eventual" should be changed to "eventually".

Page 13, notation 3, "of" should be changed to "or".

Page 6, line 15, F5e7 should be 5e7F?

Page 8, line 9, F5e6 should be 5e6F?

---

## Author Comment (AC1) · 7 Feb 2020

Author replies to anonymous Referee #3

2.) Specific comments:

1. P2-3. "illustrate the new work in this study compared to Miltenberger". Miltenberger studied relative non-extreme, small scale convective storms over a small area in the UK. There is a need to extend her analysis to a range of cases, particularly ones that can be considered truly 'extreme' in global terms. This is part of longer term effort to build-up information on the performance of bulk-microphysics scheme with two-way

coupled cloud & aerosols. The aim is to create a body of evidence which will help the NWP community to assess the value of such schemes for weather forecasting, particularly heavy rainfall in complex aerosol environments (e.g., China). We've motivated this as follows on p4: "In terms of convective storms globally, Miltenberger et al considered relatively low-intensity, small-scale rainfall events. To improve understanding of the effects of aerosols on forecasts of extremes of rainfall, there is a need to test cloud-interacting models of bulk-cloud microphysics on heavier-rainfall cases. The monsoon regions present an ideal setting for doing this because of the frequent occurrence of globally significant rainfall extremes. In this paper, we apply the CASIM microphysics to a squall line over south China which, in contrast to previously studied cases, was close to 1000-km across, produced clouds over 15 km in depth, and sustained heavy rainfall rates for a period of several days."

2. P4. L42–44. Figs 2(a,b) : contours unclear. We've changed the figure from 2x2 to 2x3-panels, and put cloud/snow & ice and rain/ graupel on separate panels (each with its own color scale).

3. Cloud-droplets size plots have been added to Figure 3. In the text: "Aerosol also affects the size of the hydrometeors (Figs 3(j-l)), with the mean-size of cloud droplets becoming larger as the aerosol concentration is reduced. The rain-drop sizes (the grey contours in Figs (j-l)) show the opposite trend, with the cleaner simulations associated with smaller rain drops (see also Fig. 4)."

4. P6. L23–25. The auto-conversion parametrization in CASIM uses the "KK"-formulation (Khairoutdinov and Koga, 2000). The parametrization depends on number-concentration and cloud-water content (not on size explicitly). Now described in the Model description section (p5): "Conversion of cloud-droplets to rain is parametrized following Khairoutdinov and Kogan (2000). Other inter-species transfers of mass and number are handled as accretion processes with bulk-collection kernels determined by the fallspeeds and collision-cross sections of the sedimenting particles."

5. P6. L27–29. "please add the initial vertical profiles of aerosol". This profile is not shown, because the initial aerosol concentrations are constant. Indeed, the depletion of aerosols near the melting level in 1s0dP is due to the mechanism that you mention (activation of droplets) .

6. The aerosol mass profile has been added to Fig.4 for the 1s0dP experiment. The panel labels have been corrected (note that two new panels have been added, in response to Review 4's comments).

Previously, the cloud-droplet number profiles ended around 1e5 -/kg because of a mistake in the plotting! Thank you for highlighting this; it has now been corrected.

7. In the original text, the value of 1/8 is just an example value, to illustrate our points regarding dependence of rain rate on aerosol. However, it was a bit confusing, so we've removed this line.

8. P11. L18. "prove that lower droplet numbers is giving rise to smaller raindrops" (as opposed to a cold-rain mechanism). We've added rain-drop size information to the longitude-time plots (Fig.3) and vertical profile plots (Figs 4(a-d). You can see from these that rain-drop size is decreasing as the cloud-droplet number decreases. In the text (p10): "[Enhancement of warm-rain processes] also predicts the simulated tendencies for rain-drop size [shown in Figs 4(a-d)]: faster warm-rain process can lead to more numerous and smaller-sized rain drops."

We agree with you that cold-rain effects cannot be ruled out; a remark about the possible effects of cold-rain processes (e.g., riming) on the reflectivity-factor histograms has been added on (p13); "Cold-rain processes could also influence th occurrence of small reflectivity values and rain rates. For example, changes in the rate-of-production of graupel particles of different sizes could influence the distribution of drop sizes after melting, and therefore contribute to the skewing of the reflectivity histograms towards smaller values of dBZ."
9. Fig. 6b. "check the lower limit of surface rain rate . . . is 0.1 mm/h". In fact, not using a lower limit of 0.1 in the model's histograms is intentional: threshold-masking the values introduces a 'normalization issue' because there would be a different total number of points in each histogram, which makes it difficult to interpret the differences. Instead, we much prefer to show the 'raw' number of counts in each histogram bin, with the caveat that the numbers below 0.1 cannot be compared to the observations.

10. Good point! This is apparently a systematic/structural error in the simulated clouds. Added to the text(p14): "It is noteworthy that the most polluted experiment (5e8F) shows the best agreement with the observations for TOA-SW flux but has relatively poor performance for radar reflectivity and surface rainfall rate. This may be because larger cloud-droplet numbers suppress drizzle, which is beneficial for stratiform clouds, and increase ice-crystal number –which is beneficial for the SW reflected from cirrus anvils– but simultaneously increase heavy-rain production in the convective-core region."

11./12. We now discuss these points further on p15–17. The main effect of two-way coupling is the novel 'clean-core' structure, as was already noted in the Conclusions. In terms of benefits for NWP: these are limited, based on the evaluations presented. We've added this as a conclusion as follows (p17): "However, despite the large changes in the microphysical structure of the squall line, e.g., the occurrence of a 'clean-core', the overall impact of coupling on model performance against the metrics considered is small, particularly when compared to the overall biases present in all the model configurations. Hence the usefulness of the additional complexity of two-way coupling for weather forecasting is still moot, and will benefit from analysis of further cases in future work."

3.) Typing errors & citations:

Checked, and corrected (where found)

---

## Author Comment (AC2) · 7 Feb 2020

Author replies for anonymous Referee #1

Specific comments:

1. Isolating the effects of giant CCN was not among the aims of this paper. We've clarified this in the Introduction, and explained more clearly the reasons underpinning our choice of model set-up (p3):

"Two-way coupling represents the minimum level of complexity in model physics required to represent depletion of aerosol during activation. We note that there exists an

lower complexity, double-moment system, in which aerosol are depleted by activation but are not recycled through clouds. It is not our intention to investigate such models here because they suffer from similar physical inconsistencies to single-moment schemes, and hence do not give a physically meaningful representation cloud-aerosol coupling. In this paper we will compare the commonly used fixed-aerosol assumption to the minimum-complexity, two-way coupling ((2), above); with the aim of understanding what new phenomena –if any– arise from consistently coupling clouds to aerosols, and whether these provide any benefits for model performance. By considering fixed-aerosol experiments with a range of aerosol concentrations, we identifying candidate mechanism for the differences between the one- and two-way coupled simulations."

We've also changed the abstract slightly to avoid giving the impression that we are seeking to isolate the effects of re-cycling from those of activation/depletion (we agree that this wasn't clearly worded before): "We focus on the effects of complexity in cloud-aerosol interactions, especially depletion and transport of aerosol material by clouds. In particular, simulations with aerosol concentrations held constant are compared with a fully cloud-aerosol-interacting system to investigate the effects of two-way coupling between aerosols and clouds on a line of organised-deep convection."

What we aim to do is compare a commonly used assumption (fixed aerosol) to the minimum complexity set-up that accounts for depletion during activation. If the aerosols are activated, then they must be recycled somehow. If activated aerosols are simply 'removed'/ 'lost', the system is not physically self-consistent, and the results would have little (if any) useful meaning. In our opinion, the only meaningful way to separate the effects of depletion during activation from re-population of interstitial aerosols is to introduce additional prognostic variables for in-cloud aerosol number concentrations. Because of the large increase in model complexity that this would involve, it is much more suitable for a separate publication. We therefore wish to argue strongly against including an investigation of these effects in this paper.

2. Fig. 2 caption has been revised.

3. corrected.

4. p9.L7–8. I've edited the text to be clearer: "The cloud-water content (Figs 4(e-h)) also peaks in 4–5-km layer indicating that condensation of liquid cloud is most active at these heights". The labels on Fig. 4 have been corrected.

5. p9.L8–10. We agree: the mass of melting snow influences the rain mass. New lines added on p10 clarify this point: "Note that this does not imply that snow is unimportant for the amount of rain. In fact, precipitating snow provides the mass flux into the melting layer from above. This is evident in the vertical profiles in Figs 4(e-h), which show that the rain-water content below the melting layer is limited by mass of snow immediately above. As the number of rain drops increases, the ratio of rain to snow increases because a larger mass of rain is needed to balance the snow-fall flux from above. In other words: in the cleaner simulations, the mass-flux from melting snow is transported by a larger number of (smaller) rain drops and a larger mass of rain resides in the column. Therefore warm-rain processes modulate the rain-drop number, and the rain-water content responds to this by increasing or decreasing so that the mass-flux of frozen precipitation from above is conserved. This process is discussed in more detail in Section 3.1.3."

6. p9.L11–12. With hindsight, this sentence was unnecessary and a bit confusing – we've removed it. (Incidentally, we meant that because auto-conversion is non-linear in the number of droplets, the production of rain is fastest at the height where droplet concentrations are lowest.)

7. The fixed aerosol number concentrations in these experiments mean that the droplet-number concentration does not vary much with height below the homogeneous freezing level. Now clarified on p9: "The cloud-droplet number profiles in the 5e7F and 5e6F are relatively uniform below the homogeneous freezing because aerosol number concentration is constant in these simulations."

8. p10. "in mixed-phase clouds .. needs a detailed analysis." We've significantly rewritten and expanded Section 3 to provide a more methodical discussion warm and mixed-phase processes. The new structure is based on discussion of 3 possible 'scenarios' of cloud aerosols interaction (explained in detail on p9): (1) warm-rain-processes dominate (2) cloud-droplet freezing dominated (3) mixed-phased-feedback dominated

We discuss the relative merits of each scenario in turn. (2) can be ruled out because it is not consistent with the simulated changes in ice- and rain numbers (see text). To some extent (2) and (3) cannot be disambiguated, because both affect rain-number in the same direction. However, we note that (2) is not consistent with the orders-of-magnitude of the changes in rain and graupel: the graupel-number changes are much too small to explain the changes in rain-drop number. Further, we've add to Fig. 4 a new experiments ("5e6_ACC") in which Na=5e6 but auto-conversion and rain-cloud accretion are both turned off (for T>-4C). In this experiment, the only possible aerosol-indirect effects are via changes in mixed-phase or ice process. The results show that 5e6F_ACC is similar to 5e7, not to 5e6. This strongly suggest the cloud-aerosol effects seen are very similar to the effects of suppressing warm-rain processes. This supports the conclusion that warm-rain processes are essential for simulating the cloud-responses seen in the full-microphysics simulations. It does not, of course, completely rule out the additional importance of mixed-phase processes, and this is noted in the revised text.

9. p10. Fig 5. Black symbols on Fig. 5. Thess symbols are for rain rates greater that 16 mm/h. The colored text-labels indicated the location of rain-rate-bin edges, i.e., the lie between the rows of colored symbols. We've clarified this in the Fig. 5 caption.

10. p12. Fig 6. Number of grid-points. An axis label has been added.

11. p15.L9–10. Agreed, the expanded Sec. 3 provides a more detailed analysis of this claim. Also the conclusions text (p16) has been modified to reflect the new analysis: "The simulations performed do not place an unambiguous constraint on aerosol effects mediated by mixed-phase processes (for example, the affects of aerosols on riming),

but experiments with direct cloud-to-rain conversions turned off suggest that warm-rain processes are at least essential to the simulated cloud responses."

I've also added a brief summary of the main model assessment results (p15): "The comparisons to observations show that forecasts with lower aerosol concentration give better predictions of histograms of hourly instantaneous rainfall rates. However, the same configurations underestimate the observed fluxes of short-wave radiation radiation in regions where cloud- and rain-water paths are large, but underestimate reflected SW fluxes from lightly precipitating cloud with low liquid-water paths."

Technical corrections :

all corrected; except for units of mm/h for rainfall amount (we believe this to be correct)

---

## Author Comment (AC3) · 7 Feb 2020

Author replies to anonymous Referee #4

Major changes:

1. "A more detailed .. discussion on the microphyical processes". We've significantly rewritten and expanded the text on p9 to provide a more methodical discussion warm and mixed-phase processes. The new structure is based on discussion of 3 possible 'scenarios' of cloud aerosols interaction: (1) warm-rain-processes dominated: more droplets implies reduced auto-conversion and fewer rain drops. (2) cloud-droplet freez-

ing dominated: more cloud-droplets leads to more ice particles (via ice nucleation); this leads to more rain drops due to melting snow as aerosol increases (3) mixed-phased-feedback dominated: the rate of riming is the dominant factor; more cloud droplets number leads to less riming (because the droplets are larger), less graupel, and fewer rain drops from melting graupel.

We discuss the relative merits of each scenario in turn. (2) can be ruled out because it is not consistent with the simulated changes in ice- and rain numbers (see text). To some extent (2) and (3) cannot be disambiguated, because both affect rain-number in the same direction. However, we note that (2) is not consistent with the orders-of-magnitude of the changes in rain and graupel: the graupel-number changes are much too small to explain the changes in rain-drop number. Further, we've added to Fig. 4 a new experiment ("5e6_ACC") in which Na=5e6 but auto-conversion and rain-cloud accretion are both turned off (for T>-4C). In this experiment, the only possible aerosol-indirect effects are via changes in mixed-phase or ice process. The results show that 5e6F_ACC is similar to 5e7, not to 5e6. This strongly suggest the cloud-aerosol effects seen are very similar to the effects of suppressing warm-rain processes. This supports the conclusion that warm-rain processes are essential for simulating the cloud-responses seen in the full-microphysics simulations. It does not, of course, completely rule out the additional importance of mixed-phase processes, and this is noted in the revised text.

Figure 5. Specifically the co-dependencies of ice-water path and rainfall rate. Firstly note that in each experiment, the surface rainfall rate will always be correlated with ice-water path, because heavier rainfall occurs underneath convective cells with larger water contents. The effects of aerosols can only be seen by stratifying the cloud properties based on rain rate (as shown by the different colors in Fig. 5). A new subsection 3.1.3 (p12) has been added specifically discussing the possible effects of ice-phase processes on rain- and ice-water paths. To paraphrase what is written there, we cannot completely rule out the role of ice-phase processes (and this is fully acknowledge

in the text), but the simulation without warm-rain process has rain- and ice-water paths that are very similar to the (more polluted) 5e8F simulations, which at least points to warm-rain processes being crucial for capturing the aerosol-effects.

p6.L14 "1s0dp has more rain, but less snow". These difference are now dealt with in the new sub-section 3.1.1 (p12): "The number of ice crystals (green [lines in Figs 4(a-d)]) is smaller in simulations with fewer cloud droplets aloft. This is consistent with fewer cloud droplets leading to less nucleation of ice, via either homogeneous freezing or heterogeneous (immersion) freezing."

p9.L16-17. "Some of the discussion of the ice phase processes may be wrong . . . there is more graupel in the low aerosol experiment . . . ". Your suggestion is that the number of rain drops may be changing in response to a change in the number of melting graupel particles. This is a possibility, but it is difficult to justify because the change in graupel numbers is much smaller than the change in rain drops. This possibility is introduced and more carefully discussed in the revised Section 3 (see "scenario 3", above. It cannot be completely discounted on the basis of the current experiments, but it struggles to explain the mismatch between graupel- and rain-number changes. Moreover, the 'no-warm-rain processes' experiment (5e6_ACC) includes the effects of changes in cloud-drop number on graupel, but does not reproduce the rain differences between the full-microphysics experiments.

p9.L14-15. "Is is hard to understand this sentence". Agreed! Moreover, I don't think the sentence was useful, I've removed it from the revised version.

p9. Rain water content and surface rainfall rate responses to aerosols. The revised Section 3 address the changes in number, condensed mass and rain-fall rate, systematically and in a more logical sequence. Number changes are dealt with first (Sec. 3.1.1). The responses of water paths then have their own subsection (3.1.2), where the reasons why rain-water path, cloud-water path and rainfall rate varying together are explained. We agree that the responses of rain-water path to aerosol are less readily understood than those of rain-drop number. The responses are explained in Sec. 3.1.2. The basic mechanism here is that larger drops fall faster so if the rain water is held constant then the precipitation rate will increase as the number of rain-drops increases. Similarly, if the rainfall rate is constant, the a larger number concentrations requires a large rain-water path to balance this precipitation flux. These mechanisms can also be put on a mathematical basis by considering mass-balance in the steady updraft. This argument is (we believe) relatively well known, but we've added short appendix (A1) given the details.

p9.L12. "the 'minimum' cloud droplet concentration." 'Minimum' referred to the lowest values of cloud-drop number attained in each column of the model the grid. However, I now think this sentence was unnecessary (and a bit confusing) –so we've removed it entirely, as part of the re-organizing in response to your comment 1. As requested, the large reduction in cloud-droplet number in 1s0dp, compared to 5e7F, is now emphasized in the abstract & conclusions.

2. Model description. (Section 2.1.) We've now described the choice of immersion freezing parametrization in more detail on p5: "This fraction is a function only of temperature and is independent of the number of interstitial aerosol particles. Ice-crystal number concentration can be indirectly affected by the number of aerosol particles, because the number of cloud droplets can affect the number of ice crystals." We've now described the ice-cloud responses seen in vertical profile (Fig. 4) on p9: "The number of ice crystals (green) is smaller in simulations with fewer cloud droplets aloft. This is consistent with fewer cloud droplets leading to less nucleation of ice, via either homogeneous freezing or heterogeneous (immersion) freezing."

The choice of aerosol concentrations is now discussed on p5: "For the fixed-aerosol experiments we consider reductions $N_a$ in decades, from approximately 500/cmˆ3. This range is selected to span the range of concentrations generated in 1s0dP. For reference, some of the plots also include an unrealistic, 'limiting' case with $N_a=50ˆ5$ -/kg."

We agree 5e5F is unrealistically clean, we still think it's worth including as a 'limiting case'.

Regarding the rainfall rate variations: we agree the sensitivity of these to aerosol perturbations is not large across the range considered. In fact, this a conclusion of this study: rainfall intensity and area change, but in a way that keeps rainfall amount (rain rate) approximately constant.

3. Abstract. We've modified the abstract to be more specific about the paper's conclusions. The new part reads: "It is shown that in-cloud processing of aerosols can change the vertical structure of the storm by using up aerosols within the core of line, thereby maintaining a relatively clean environment which propagates with the heaviest rainfall. This induces changes in the statistics of surface rainfall, with a cleaner environment being associated with less intense but more frequent rainfall. These effects are shown to be related to a shortening of the timescale for converting cloud-droplets to rain as the aerosol-number concentration is decreased. The simulations are compared to satellite-derived estimates of surface rainfall, condensed-water path and the outgoing flux of short-wave radiation. Simulations with fewer aerosol particles out-perform the more polluted simulations for surface rainfall, but give poorer representations of top-of-atmosphere radiation."

Minors changes:

Figs 1–3; text size. I'm not in favour of increasing this; it's a trade off between text size and picture size. I think the text is readable.

Fig. 7. the 20pc criteria is only used to demarcate ice clouds on the plot – it is not used in a statistical analysis.

Fig. 8. labels corrected.

Various typos corrected.